# The Ketogenic Diet: Is It an Answer for Sarcopenic Obesity?

**DOI:** 10.3390/nu14030620

**Published:** 2022-01-30

**Authors:** Zahra Ilyas, Simone Perna, Tariq A. Alalwan, Muhammad Nauman Zahid, Daniele Spadaccini, Clara Gasparri, Gabriella Peroni, Alessandro Faragli, Alessio Alogna, Edoardo La Porta, Ali Ali Redha, Massimo Negro, Giuseppe Cerullo, Giuseppe D’Antona, Mariangela Rondanelli

**Affiliations:** 1Department of Laboratory, Bahrain Specialist Hospital, Juffair P.O. Box 10588, Bahrain; 2Department of Biology, College of Science, Sakhir Campus, University of Bahrain, Zallaq P.O. Box 32038, Bahrain; simoneperna@hotmail.it (S.P.); talalwan@uob.edu.bh (T.A.A.); nzahid@uob.edu.bh (M.N.Z.); 3Endocrinology and Nutrition Unit, Azienda di Servizi alla Persona “Istituto Santa Margherita”, University of Pavia, 27100 Pavia, Italy; daniele.spadaccini01@universitadipavia.it (D.S.); clara.gasparri01@universitadipavia.it (C.G.); gabriella.peroni01@universitadipavia.it (G.P.); 4Department of Internal Medicine/Cardiology, Deutsches Herzzentrum Berlin, 13353 Berlin, Germany; alessandro.faragli01@gmail.com; 5Department of Internal Medicine and Cardiology, Campus Virchow-Klinikum, Charité—Universitätsmedizin Berlin, 13353 Berlin, Germany; alessio.alogna@charite.de; 6DZHK (German Centre for Cardiovascular Research), Partner Site Berlin, 10785 Berlin, Germany; 7Berlin Institute of Health (BIH), 10178 Berlin, Germany; 8Department of Cardionephrology, Istituto Clinico Ligure Di Alta Specialità (ICLAS), GVM Care and Research, 16035 Rapallo, Italy; edoardo01laporta@gmail.com; 9Department of Internal Medicine (DiMi), University of Genova, 16121 Genova, Italy; 10Department of Chemistry, College of Science, Sakhir Campus, University of Bahrain, Zallaq P.O. Box 32038, Bahrain; ali96chem@gmail.com; 11Chemistry Department, School of Science, Loughborough University, Loughborough LE11 3TU, UK; 12CRIAMS-Sport Medicine Centre, 27058 Voghera, Italy; massimo.negro@unipv.it (M.N.); gdantona@unipv.it (G.D.); 13Department of Movement and Wellbeing Sciences, University of Naples “Parthenope”, 80133 Napoli, Italy; giuseppe.cerullo@uniparthenope.it; 14Department of Public Health, Experimental and Forensic Medicine, University of Pavia, 27100 Pavia, Italy; mariangela.rondanelli@unipv.it; 15IRCCS Mondino Foundation, 27100 Pavia, Italy

**Keywords:** sarcopenia, ketogenic diet, gut microbiota, visceral adipose tissue (VAT), cytokine, fatty liver, physical inactivity

## Abstract

This review aims to define the effectiveness of the ketogenic diet (KD) for the management of sarcopenic obesity. As the combination of sarcopenia and obesity appears to have multiple negative metabolic effects, this narrative review discusses the effects of the ketogenic diet as a possible synergic intervention to decrease visceral adipose tissue (VAT) and fatty infiltration of the liver as well as modulate and improve the gut microbiota, inflammation and body composition. The results of this review support the evidence that the KD improves metabolic health and expands adipose tissue γδ T cells that are important for glycaemia control during obesity. The KD is also a therapeutic option for individuals with sarcopenic obesity due to its positive effect on VAT, adipose tissue, cytokines such as blood biochemistry, gut microbiota, and body composition. However, the long-term effect of a KD on these outcomes requires further investigations before general recommendations can be made.

## 1. Introduction

### Sarcopenic Obesity and Its Impact on Health Outcomes

The link between sarcopenia and obesity is known as the sarcopenic obesity (SO) syndrome. Due to the combination metabolic burden of reduced muscle mass (sarcopenia) and excess adiposity, it is a substantial risk factor (obesity) [1]. As recently proposed by Barazzini et al. [2], the term ‘sarcopenic obesity’ has been used to describe obesity with reduced skeletal muscle function and mass. However, its application is mostly limited to the elderly patient group, and there is still a lack of agreement on its description and diagnostic criteria. SO is sometimes characterized as obesity combined with sarcopenia, which can occur in older people, those with type 2 diabetic mellitus (T2DM) or chronic obstructive pulmonary disease (COPD), and obese patients with malignant illnesses who are losing weight [3]. Although sarcopenic obesity is the co-existence of sarcopenia and obesity in a person, the fundamental difficulties in molecular connections between skeletal muscle and adipose tissue should be investigated further [4,5].

Excessive calorie consumption, inactivity, low-grade inflammation, insulin resistance, and hormonal changes can all contribute to the development of SO [6]. The main criteria for defining visceral obesity in association with obesity, according to the World Health Organization, are a body mass index BMI of 30 kg/m^2^ and a waist circumference of >102 cm in men and >88 cm in women [7]. The development of visceral adiposity is linked to a number of inflammatory mechanisms. Obesity activates macrophages, mast cells, and T lymphocytes, resulting in low-grade inflammation and tumor necrosis factor tumor necrosis factor (TNF), leptin, and growth hormone secretion [8]. These hidden alterations cause insulin resistance (IR), which is exacerbated by muscle dysmetabolism [9], favoring fat mass accumulation and muscle mass loss [7,10]. As a result of the energy imbalance caused by high calorie consumption, patients with SO experience inflammation, which lowers muscle mass. Positive benefits of exercise, the role of gut hormonal systems and microbiota metabolism, and the role of brain regulation of physical and skeletal muscle activities are all key mediators [2].

Due to the possible induction of a state of subclinical systemic inflammation [11], the risk of obesity-related diseases is closely linked to visceral adipose tissue (VAT). VAT is associated with an increased risk of developing chronic diseases such as cardiovascular disease, T2DM, stroke, and musculoskeletal disorders [12]. The osteosarcopenic obesity (OSO) syndrome is defined as the coexistence of bone tissue degeneration, sarcopenia, and fat accumulation, which results in decreased physical function and systemic metabolic dysregulation [13]. According to a study by Dimitri et al. [14], the effect of adiposity on bone varies depending on the location of fat deposition. Visceral fat acts as a pathogenic fat depot, whereas subcutaneous fat acts as a protective fat depot. Perna et al. [15] added to the evidence by finding a link between adiposity and bone mineral density, which they explained by enhanced aromatization of androgens to weak estrogens in subcutaneous adipose tissue. Poor nutritional condition and inflammation are the causes of OSO, according to JafariNasabian et al. [16], while fractures are the result of OSO. According to Ilich et al. [17], the incidences of sarcopenia, SO, and OSO were 31.5%, 5.1%, and 4.1%, respectively, in a recent study. Sarcopenia is commonly seen in chronic conditions such as malignant tumors, persistent rheumatoid arthritis, and cirrhosis, to name a few. When these disorders are accompanied by sarcopenia, the risk of unfavorable outcomes rises, and the prognosis worsens. Visceral obesity is well-known for causing IR, which increases the risk of cancer and cardiovascular problems. Thus, skeletal muscle mass and visceral fat mass can influence disease pathogenesis, with different consequences depending on the condition [18,19].

The goal of this study is to determine how effective the ketogenic diet (KD) is at managing SO. Since the combination of sarcopenia and obesity appears to have multiple negative metabolic effects, this narrative review examines the effect of KD as a possible synergic intervention to modulate and improve the gut microbiota and to reduce VAT by cytokine secretion, fatty infiltration of the liver, and the sense of chronic fatigue in people with SO.

## 2. Materials and Methods

The narrative evaluation was carried out according to Egger et al. [20]. Randomized controlled trials, cross-sectional studies, and observational studies on the effects of a KD on sarcopenia and obesity were suitable for the systematic review. The following medical subject headings were used in the search strategy: (sarcopenic obesity OR sarcopenia OR muscle) AND (keto-genic diet) AND adults. The articles were retrieved from “PubMed” and “Scopus” during the initial search.

## 3. Results

### 3.1. The Ketogenic Diet

A KD’s macronutrient profile is composed of 55 to 60% lipids, 30 to 35% protein, and 5 to 10% carbohydrates. This causes nutritional ketosis, in which fatty acids undergo partial beta-oxidation to produce ketone bodies, which are then used as a source of energy. Ketone bodies are used to replace glucose as a source of energy in most tissues throughout time. Similar to extended fasting, subcutaneous adipose cells re-release fatty acids from depots, which are partially beta-oxidized to produce ketone bodies [21]. KD appears to increase survival by modulating mammalian target of rapamycin (mTOR), as evidenced by multiple studies in rats. The increase in lifetime has been linked to changes in mitochondrial quantity and quality in both skeletal muscle and the liver, and this could occur without a large change in oxidation status [22,23,24,25].

Merra et al. [26] found that a KD was highly effective in reducing body weight without causing a loss of lean body mass, thereby preventing sarcopenia, in a human investigation. Rauch et al. [27] conducted a similar study on 26 college-aged, resistance-trained men and found that those who ate a KD had lower fat mass than those who ate a standard western diet.

By increasing lean mass and decreasing inflammation and oxidation [28], ketone bodies, primarily beta-hydroxybutyrate (BHB), cause physical and chemical changes such as satiety, changes in body composition, and a decrease in hormone-dependent hunger by causing energy alterations at the mitochondrial level. In addition, Benlloch et al. [29] investigated whether the satiating impact of a KD may enhance body composition and oxidation levels in patients with multiple sclerosis. It was discovered that there was a considerable rise in satiety perception at lunch and dinner, as well as BHB levels in the blood. With equal amounts of ghrelin, hunger perception reduced dramatically at lunch and dinner.

Low muscle mass and quality, as well as visceral adiposity and sarcopenic visceral obesity, are all linked to death and recurrence following pancreatic cancer resection. Ketone bodies are crucial alternative fuels that allow humans to survive during periods of glucose deficiency caused by famine or extended exercise [30].

It has also been highlighted that KD plays a role in treating nonalcoholic fatty liver disease (NAFLD) by lowering hunger and concurrently decreasing carbohydrate consumption, owing to two separate pathways, which include the general reduction of body weight and the modulation of insulin levels. At the same time, since KD promotes fat oxidation, an increase in diet fat percentage does not result in an increase in liver fat [31].

### 3.2. The Effect of KD in Reduction of Inflammation

Cytokines are tiny proteins that cells produce and have a specific influence on cell interactions and communication. Cytokines can have an autocrine or paracrine effect on the cells that secrete them, as well as on neighboring cells and, in certain cases, distant cells (endocrine action). Pro-inflammatory and anti-inflammatory cytokines are both present [32]. Adipocytes and adipose tissue leukocytes release proinflammatory cytokines and leptin, which have a direct catabolic effect on muscle cells and also act indirectly via IR, reducing the anabolic effect of insulin on muscle cells [33].

Lim et al. [32] investigated the links between the chemokine monocyte chemoattractant protein-1 (MCP-1) and sarcopenia, obesity, and the SO characteristics in groups of older persons. Of the 143%, 25.2% were normal, 15.4% were sarcopenic, 48.3% were obese, and 11.2% were SO. MCP-1 levels were shown to be significantly higher in obese and SO patients, validating the idea of chronic inflammation caused by excess adiposity. As a result, inflammatory cytokines appear to have a key role in the pathogenesis of obesity.

Roubenoff et al. [34] found that aging is linked to increased production of catabolic cytokines, decreased circulating levels of insulin-like growth factor-1 (IGF-1), and sarcopenia acceleration (loss of muscle with age). The findings showed that higher levels of the catabolic cytokines TNF-alpha and interleukin-6 (IL-6), are linked to increased mortality in community-dwelling elderly adults, whereas insulin like growth factor-1 (IGF-1) levels have the opposite effect [35,36]. Chronic use of KD, on the other hand, causes the host to synthesize adenosine triphosphate (ATP) utilizing -hydroxybutyrate and reduces NOD-like receptor P3 (NLRP3) mediated inflammation. Similar studies indicated that the KD reduced fat mass in numerous fat depots, including visceral fat [37,38]. There-fore these findings are not applicable to all circumstances.

In diabetic and insulin-resistant conditions, elevated levels of TNF, IL-6, and IL-8 have all been recorded. Through autocrine and paracrine signaling, obesity-induced alterations in skeletal muscle, adipose tissue, and the liver cause localized inflammation and IR. IR in distant tissues is caused by endocrine-mediated interaction between insulin target tissues. The net result of these alterations is systemic inflammation and IR [39].

### 3.3. Effect of KD on Visceral Adipose Tissue in Sarcopenic Obesity

Visceral obesity is also linked to metabolic syndrome and insulin resistance indica-tors. VAT synthesizes and secretes various hormones to communicate with other central and peripheral organs as a metabolically active organ. Adipokines are hormones that are directly linked to metabolic balance, emphasizing the importance of adipose tissue in the regulation of energy homeostasis [40]. For different malignancies, visceral fat gain and muscle loss have been recognized as unfavorable prognostic variables [41].

Low muscle mass and quality, as well as visceral adiposity and sarcopenic visceral obesity, are all linked to death and recurrence after pancreatic cancer resection [30]. Ketone bodies are crucial alternative fuels that allow humans to survive in periods of glucose deficiency caused by fasting or extended activity. Chronic use of KD, on the other hand, causes the host to generate ATP utilizing -hydroxybutyrate and reduces NLRP3-mediated inflammation [42,43].

Cunha et al. [44] found weight loss in the very low-calorie ketogenic (VLCK) diet group with VAT reduction in a trial of 46 patients. Another investigation on adiposity parameters and orexin-A serum profile by Valenzano et al. [45] found similar results, namely that the VLCK diet reduced VAT, improving adiposity and blood biochemistry, while Orexin-A levels considerably rose after dietary treatment.

A recent study suggests that the mechanism by which KD causes visceral fat loss could be related to the satiety-increasing effect of higher dietary protein, and that this effect could be regulated by appetite-mediating hormones such as leptin, which is produced by adipose cells and is involved in the regulation of energy balance and fat storage by suppressing hunger [46].

Bertoli et al. [35] looked at how a 12-week KD altered adipose tissue activity indicators, VAT, and subcutaneous fat in children with glucose transporter 1 deficiency syndrome. Despite being a high-fat diet, the results showed lower fasting insulin levels, implying that KD had no effect on abdominal fat distribution over a short period of time.

### 3.4. Effect of KD on Nonalcoholic Fatty Liver Disease (NAFLD) in Sarcopenic Obesity

NAFLD is a condition in which fat accumulates in the liver and is one of the most common types of chronic liver disease in industrialized countries [47]. In Western countries, the prevalence of NAFLD is estimated to be 20–30% in the general population; in obese populations, this increases to 57.5–74% [47,48]. Insulin activity affects both the liver and the muscle. IR is thought to play a role in the pathogenesis of both NAFLD and sarcopenia. NAFLD is now recognized as both a cause and a result of IR. Hepatic steatosis is the initial stage of NAFLD, defined by intrahepatic triglyceride (IHTG) values greater than 55 mg/g liver (5.5%) or when more than 5% of hepatocytes show histological signs of triglyceride storage [48].

Petta et al. [49] reported the presence of fibrosis in sarcopenic patients to the extent that the prevalence of sarcopenia was linearly associated with the severity of fibrosis (OR 2.36, CI 1.16–4.77, *p* = 0.01). Furthermore, the presence of sarcopenia was seen in 48.3% of patients with severe fibrosis compared to 20.4% in the mild fibrosis group (i.e., a fibrosis grade less than or equal to two (*p* < 0.01)).

Obesity and IR, which are both common in NAFLD, play a critical role in the pathogenesis of liver damage and the advancement of cirrhosis and end-stage complications [50]. Other potential disease progression mechanisms (vitamin D insufficiency, hyperuricemia, industrial fructose intake, menopausal status, etc.) have been hypothesized, and they are typically shared by NAFLD, obesity, and IR, further complicating the complicated interplay between NAFLD and metabolic dysfunction [51]. Tendler et al. [52] found that a low-carbohydrate, KD reduced mean weight by −12.8 kg and improved histologic fatty liver disease after six months in research on the effect of low-carbohydrate, KD on NAFLD. Watanabe et al. [31,53] recently published a study that found that a low-carbohydrate diet (LCD) improves liver fat metabolism in obese NAFLD patients.

Browning et al. [54] used a carbohydrate-restricted (20 g/d) or calorie-restricted (1200–1500 kcal/d) diet on NAFLD individuals for two weeks vs. an isocaloric formula as a placebo in a clinical investigation. The weight reduction was observed to be similar in both groups (−4.0 1.5 kg in the calorie-restricted group and −4.6 1.5 kg in the carbohydrate-restricted group). Although liver triglycerides reduced with weight loss, they decreased much more in carbohydrate-restricted patients (−55 14%) than in calorie-restricted subjects (−28 23%).

Mice lose weight, develop ketosis, and produce hepatic gene expression patterns that suggest reduced de novo lipogenesis and increased fatty acid oxidation when fed a micronutrient supplemented KD that is high in fat (93.3% kcal), low in carbohydrate (1.8%), and low in protein (4.7%) [55,56].

### 3.5. Gut Microbiota in Sarcopenic Patients and Effect of Ketogenic Diet

There are up to 1014 bacteria, viruses, fungi, protozoa, and Archaea in the human gut microbiota. This gene pool has been predicted to be 150 times larger than the host’s, weighing between 175 g and 1.5 kg [57]. Early childhood influences the composition of the gut microbiota, which is influenced by geographical factors, the method of delivery (vaginal or cesarean), breastfeeding, weaning age, antibiotic exposure, and dietary regimens [58,59]. Even though two phyla (Bacteroidetes and Firmicutes) account for up to 99% of species, the healthy adult gut microbiota contains bacteria from ten phyla. Bacteroidetes’ average relative abundance is generally inversely proportional to Firmicutes’, and vice versa [60].

According to studies, the gut microbiota’s resilience declines around the age of 65, making its overall composition more sensitive to lifestyle changes, pharmacological therapy such as antibiotics, and disease [61]. Aging is linked to decreased microbiota richness, higher inter-individual variability, and pathobiont overrepresentation. Given that changes in the quality, quantity, and function of gut microbiota with age may contribute to chronic inflammation and anabolic resistance, its role in the development of aging sarcopenia should be explored. Furthermore, therapies that improve microbiome expression and function may help to reduce age-related decreased muscle performance and the associated negative clinical outcomes [62].

Rondanelli et al. [63] consolidate the present knowledge regarding the human microbiota in the elderly and the effects of probiotics in the elderly population in a systematic review. The results show that, when compared to the adult population, elderly people have a lower diversity of microbiota, with lower numbers of Firmicutes, Bifidobacteria, Clostridium cluster XIV, Faecalibacterium Prausnitzii, Blautia coccoides-Eubacterium rectal, and a higher presence of Enterobacteriaceae and Bacteroidetes, and a higher presence of Enterobacteriacea It was concluded that differences in the intestinal microbiota of the elderly may not be caused solely by aging, but may be linked to a decline in general health, malnutrition, and an increased need for medication, such as antibiotics and nonsteroidal anti-inflammatory drugs, all of which are common in the elderly [64].

The age-related changes in gut microbiota composition have been found to promote intestinal mucosa permeability in experimental models of aging. This phenomenon causes increased systemic absorption of bacterial products, such as lipopolysaccharide, which activates the inflammatory response and leads to higher levels of pro-inflammatory cytokines including IL-6 and TNF-alpha in the bloodstream [65].

Newell et al. [66] found that feeding a KD to juvenile male C57BL/6 (B6) and BTBR mice for 10–14 days exhibited an antimicrobial-like effect by considerably lowering overall host bacterial abundance in cecal and fecal waste. KD was also shown to reverse high *Akkermansia muciniphila* concentration in BTBR animals’ cecal and fecal matter. In two seizure mouse models, KD affects the gut microbiome. Seizure protection is conferred through changes in the microbiota, which are both essential and sufficient. KD decreased gut bacterial alpha diversity in mice while increasing *A. muciniphila* and Parabacteroides relative abundance [67].

KD also raised the relative abundance of putatively beneficial gut microbiota (*Akkermansia muciniphila* and *Lactobacillus*) while decreasing the relative abundance of pu-tatively pro-inflammatory taxa (*Akkermansia muciniphila* and *Lactobacillus*) (*Desulfovibrio* and *Turicibacter*). KD also decreased blood glucose levels and body weight while increasing blood ketone levels, which could be linked to changes in the gut microbiome [68]. The LCD causes fast changes in the gut microbiota, which raises circulating folate levels and upregulates gene expression in the liver involved in folate-dependent one-carbon metabolism. Increases in folate-dependent one-carbon metabolism gene expression in the liver appear to be the added value of LCD. Since folate mediates the conversion of sarcosine to glycine, circulating sarcosine may increase in the presence of a folate shortage. In mouse fibroblasts, sarcosine stimulates autophagy in a dose-dependent manner, and changes in myocyte quality control mechanisms (including autophagy) may contribute to sarcopenia [59].

### 3.6. The Effect of KD on Physical Performance in Sarcopenic Obesity

Individuals who resistance train recreationally have been interested in dietary approaches that aid body fat loss while maintaining or leading to an increase in muscle mass. As a result, there is substantial evidence that increased protein diets can improve muscle mass increases while having little effect on body fat [59]. Recent research in rats [69], as well as prior findings in humans [59], suggests that following a low-carbohydrate, moderate-protein, high-fat KD can help sustain muscle mass growth while also reducing adiposity. As a result, the KD is advocated as an obesity-fighting therapy.

After a 12-week KD, obese adults lost weight, improved physical performance, cognitive function, eating behavior, and metabolic profile, according to a study conducted by Mohorko et al. [70]. The study found a significant reduction in appetite and body weight (men 18 9 kg vs. women −11 3 kg; *p* = 0.001), as well as enhanced physical performance (*p* = 0.001). Similarly, after a dietary carbohydrate restriction intervention experiment, Anguah et al. [71] investigated changes in food cravings and eating behavior. The researchers expected that even after a brief (four-week) period of low-carbohydrate restriction, decreases in food cravings and improved eating habits would be noticeable. In adult participants, dietary constraint was shown to be 102% higher, whereas disinhibition and hunger scores were lowered (17% and 22%, respectively).

Gregory et al. [72] looked at how a 6-week low-carbohydrate KD and CrossFit program affected body composition and performance. In comparison to the control group, the low-carbohydrate KD group lost weight, had a lower BMI, and had a lower % body fat. The findings suggest that combining an low-carbohydrate KD with six weeks of CrossFit exercise can result in significant reductions in percent body fat, fat mass, weight, and BMI while preserving lean body mass and performance.

## 4. Conclusions

The KD is commonly utilized as an obesity-fighting technique, and it has been demonstrated to be successful in both human and animal models. Despite the fact that KDs are successful for weight loss, the relationship between biochemical, physiological, and psychological alterations is still completely unknown. Sarcopenia is related with an increase in visceral fat and a progressive loss of muscle mass as people get older. Furthermore, nutritional composition appears to play an essential influence in determining the makeup of the gut microbiota, which has been linked to sarcopenia pathogenesis. Finally, ketogenic diets with extremely low carbohydrate content have been demonstrated to lower VAT while improving adiposity and blood chemistry markers.

## Data Availability

Not applicable.

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
