# Peer review of "The Ketogenic Diet: Is It an Answer for Sarcopenic Obesity?"

_nutrients, 2022, doi:10.3390/nu14030620_

Round 1

Reviewer 1 Report

I believe this manuscript has great potential but unfortunately is not ready for publication at this point. Please see the attached document for my suggested revisions. 

Author Response

All changes have been made in the original manuscript. This file contains the answers to your question. 

Reviewer 2 Report

The authors have well addresses my concerns and have significantly improved the manuscript. I do not have further comments/concerns. I congratulate the authors for nice work.

Author Response

Thank you for your apprecitation 

This manuscript is a resubmission of an earlier submission. The following is a list of the peer review reports and author responses from that submission.

Round 1

Reviewer 1 Report

The aim of the authors is to illustrate any reasons why ketogenic diets (KDs) maybe appropriate therapeutic strategies for sarcopenic obesity. The topic is of high interest, but the manuscript raises several concerns, deriving from an overall lack of precise definition of sarcopenic obesity and ketogenic diet.

In particular:

The abstract is confusing. I would highly recommend to make it structured.

The manuscript lacks of an "aims" and a "method" section

The definition of sarcopenia and sarcopenic obesity is not univocal and there is a lack of common scientific criteria (10.1016/j.clnu.2018.04.018, 10.1016/j.clnu.2019.11.024). I would highly recommend to clarify this important point and to better schematize the most important features of sarcopenic obesity. Maybe a figure could be an option. At this time, the relative paragraph is confusing and misleading (i.e line 68 "Sarcopenic obesity is define as central adipose tissues" is a superficial and incomplete definition)

Similarly, ketogenic diets represents a category of different dietary interventions ( https://doi.org/10.1111/obr.13024)  
Again, the definition provided by the authors is incomplete (line 95).

In the paragraph related to fatty liver, several evidence are missing (10.1016/j.cmet.2018.01.005,  https://doi.org/10.1111/obr.13024, 10.3390/nu12072141.)

I would recommend to improve the readability of the manuscript by adding table for each analysis (KD-nafld, KD-inflammation etc).

Overall, the scope of the manuscript is interesting, but in order to be suitable for publication it should be revised with clear objectives, methods and definions.

Author Response

please, see the attachment

Reviewer 2 Report

I would like to thank the authors for their manuscript. I anticipate it being a valuable contribution to the literature. 

Abstract

  • Line 37-38 state that a KD reduces the resilience of the gut microbiota (which may be detrimental) yet concludes that a KD is beneficial. This needs to be explained.
  • Line 39-41 state that γδ T cells and anti-inflammatory cytokines are reduced (which is likely detrimental) yet conclude a KD is beneficial. This needs to be explained
  • Line 46 remove “such individuals”

Introduction

  • Line 63-64 this sentence is unclear and should be revised. Is “muscle metabolism” supposed to be “muscle dysmetabolism”?
  • Line 68 “define” should be “defined”. Sarcopenic obesity is not defined as central adipose tissue
  • Line 82 “OS” should be “SO”

The Ketogenic Diet

  • Line 95-96 be careful with this wording. The macronutrient balance is not what determines nutritional ketosis (the insulin state of the body determines it), although it may be effective at inducing it
  • Line 97-98 revise this sentence. It has multiple grammatical errors and potentially inaccuracies
  • Line 103-104 citation needed
  • Line 110-111 remove “reduction”
  • Line 112-113 “presented” should be “developed”. Also a systematic review (which there are several of) would provide better support than this study
  • Line 117 “The” or “A” should precede “Ketogenic diet”

Gut microbiota in Sarcopenic Patients and Effect of Ketogenic Diet

  • Line 132-133 use a apposition commas or an Oxford comma to more clearly delineate antibiotic and disease
  • Line 143 “summarize” should be “summarizes”
  • Line 150-151 use commas to delineate “general state of health”, “malnutrition”, and “increased need for medication”
  • Line 158 “A” should precede “study”
  • Line 159 should read “associated with a poor prognosis”. Also this discussion on colon cancer seems unrelated to the surrounding text
  • Line 169-170 reference 34 is cited but I believe it is meant to be a difference reference. Also it is a run-on sentence and should be broken into 2 sentences (one of which could be combine with the following sentence)
  • Line 175-176 states a high fat diet produces deleterious effects. This needs to be better connected/transitioned with the preceding sentences discussing use of a KD in epilepsy and the subsequent sentences discussing the beneficial effects of a KD
  • This section should be divided into multiple paragraphs

Effect of KD on Visceral adipose tissue (VAT) in Sarcopenic Obesity

  • Line 188-190 reference 41 does not support the author’s statement. I believe it is meant to be reference 42
  • Line 191-194 reference 42 does not support the author’s statement
  • Line 194 “A” should precede “study”
  • Line 200-201 states that a KD expands adipose tissue γδ T cells but just prior to this states that a long term KD depletes γδ T cells. This needs to be better explained. My understanding of the study is that the diets were not properly matched for protein so the KD group overate in the long term, which resulted in excessive fat gain and subsequently depletion of γδ T cells
  • Line 202 “VLCK” should be written out and the abbreviation put in parentheses
  • Line 206 “A” should precede “study”
  • Line 207 “VLCKD” should be written out and the abbreviation put in parentheses

The Effect of KD in reduction of inflammation

  • Line 217 “A” should precede “study”
  • Line 218 Chong is senior author, but Lim is first author
  • Line 229-231 these two sentences seem out of place here and better placed in “The Ketogenic Diet” section
  • Line 232 “A” should precede “study”
  • Line 235-238 has multiple grammatical errors that need to be corrected

Effect of KD on Nonalcoholic fatty liver disease (NAFLD) in sarcopenic obesity

  • Line 246-247 “whereas NAFLD… insulin resistance” should be a standalone sentence so as to not make a run-on sentence
  • Line 253 “A” should precede “study”
  • Line 256 should read “sarcopenia associated with” as “serve” should be “severe”
  • Line 258 needs an end-parenthesis
  • Line 259 “high prevalence of metabolic disorders and advanced liver disease” does not grammatically fit here and should be moved (or removed)

The effect of KD on physical performance and fatigue in sarcopenic obesity

  • Line 271 Period should be removed from heading
  • Line 274 is correct that high protein diets can improve muscle mass without affecting body fat. A randomized control trial is cited as “overwhelming support”. A systematic review would be more appropriate to claim “overwhelming support” or the language could be changed to “significant support” or something similar
  • Line 278 “A” should precede “study”
  • Line 281 remove “was”
  • Line 294 should read “under these dietary conditions”
  • This section would benefit from the addition of recent systematic reviews rather than relying on only small studies, for example https://pubmed.ncbi.nlm.nih.gov/32865567/

Conclusion

  • Line 302 “had” should be “has”
  • Line 305-306 “Since ageing… muscle mass” is not a complete sentence

Reviewer 3 Report

Please see the attached file for my comments. Topic got me excited but the content is disappointing.

Round 2

Reviewer 2 Report

Overall: the manuscript has a great premise but seems lacking in its delivery. There are times where it discussed seemingly off topic things yet is missing a lot of potential content related to the ketogenic diet affecting body composition. There is at least one factual inaccuracy and multiple instances of the contracting statements. There are also frequent grammatical mistakes. I believe this manuscript can be a valuable contribution to the literature, so I have voted to reconsider after major revision. However, if the bolded comments are not corrected with the next revision, I will vote to reject, so I would recommend the authors withdraw their submission to have time to significantly revise their manuscript and resubmit.

Title: Recommend changing to “The Ketogenic Diet: Is it an Answer for Sarcopenic Obesity?”

Section order: recommend changing to Visceral adiposity -> NAFLD -> Gut microbiota -> Inflammation -> Physical performance

Lines 37-41 “The results of this review show that the KD reduces the resilience of the gut microbiota, could be a useful intervention for the management of sarcopenic obesity. It also reduces the growth of Bacteroidetes and Firmicutes, altering microbiome biodiversity. The KD depletes adipose-resident γδ T cells. Long-term treatment with the KD reduces anti-inflammatory cytokine secretion by reducing interleukin-6 and tumor necrosis factor-α and decreases abdominal fat distribution despite being a high-fat diet.” Please see comments for lines 199-210 and lines 227-233. In brief, it reads that the authors are contradicting themselves in regard to gut microbiota as well as γδ T cells.

Line 46 “require” should be “requires”

Line 73: “Sarcopenic obesity is defined as central adipose tissues”- this in inaccurate. Please correct

Line 74-75: remove “and”. Remove “also at stem cell level”. Change “important areas of include mediators of” to “important mediators include”

Line 76-77: remove “and their nutritional regulation in altering skeletal muscle homeostasis with potential muscle-catabolic systemic alterations”. Add “and” before “the

Line 98-104: Why is body recomposition (decrease in fat mass with concomitant maintenance or increase in lean mass) not a topic of this review? That seems central to answering sarcopenic obesity and there are numerous studies on it. I understand the visceral adiposity and physical performance relate to this, but it should be more directly addressed. Also the “sense of chronic fatigue” is never addressed in the section on physical performance so I would change it to “physical performance”

100-101: Change “, and” to “as well as”. Remove “by cytokine secretion”

Line 114: “adequate protein” is not defined by a percentage of calories in the diet

Line 115: “this macronutrient balance 114 effectively induces nutritional ketosis”

Line 115-117: “The process takes place in liver where carbohydrates are converts fats into fatty acids, results in the production of ketone bodies (KB)” – this is inaccurate. Please correct

Line 121: the study done by de Cabo does not have a citation in the references. Also please clarify if “calorie-restricted mice, animals fed with lard had 40%” is one group or two groups

Line 124-127: “In addition, a study done by Parry et al. suggested that the KD may influence oxidative stress by affecting mitochondrial quantity and quality, and perhaps lengthen lifespan.”  - the abstract of the study states that it does not affect oxidative stress. Also stating “may influence” contradicts line 131-134 which states “without altering oxidative stress markers.”

Line 135 “Ketogenic” should not be capitalized

Line 140-147: this new content has multiple grammatical errors that need to be revised. It is also missing multiple citations. Most of this seems like it should be in the NAFLD section rather than here. “Ketosis itself may be involved in NAFLD pathogenesis” is never further explained in the paper and is contradictory to the premise. The ketone ester part could be better placed in the physical performance section.

Lines 187-192: the sentences on colorectal cancer seem tangential. I think the section would be clearer and flow better without them

Line 192-196: mice studies are interesting, but it would be better to cite human data on gut microbiota changes with a ketogenic diet as mice and humans may differ. One example would be https://pubmed.ncbi.nlm.nih.gov/24336217/

Lines 197-205: the sentences on seizures and epilepsy seem tangential. I think the section would be clearer and flow better without them. I would instead focus on how gut microbiota changes relate to body composition as there are only 3 lines on it (207-210)

Line 199-210: the authors state “the KD reduces gut bacterial alpha diversity” and a “high fat diet induces large alterations in microbiota producing deleterious effects on gut health”. Yet also state KD “increased relative abundance of putatively beneficial gut microbiota and reduced that of putatively pro-inflammatory taxa”. If the authors are trying to say that the decreased diversity results in decreased deleterious microbiota, this is unclear. It currently reads to me that the authors are saying contradicting statements.

Line 212: “Visceral adipose tissue (VAT)”- abbreviation has already been defined earlier. Section titles should have each major word capitalized (ie adipose tissue)

Line 224: “ketogenic diet (KD)” – abbreviation has already been defined earlier

Line 227: “Study” should not be capitalized

Line 227-233: “long-term ad libitum KD feeding in mice causes obesity, impairs metabolic health, and depletes the adipose-resident γδ 228 T cells” contradicts “KD initially improves metabolic health and expands adipose tissue γδ T cells.” I understand that the authors are suggesting γδ T cells link fatty acid use to adipose inflammation, but this does not explain why there is a difference in γδ T cells amount over time on a ketogenic diet. The authors need to explain why the KD group initially increased γδ T cells and then later decreased them

Line 235-244: “VLCK” abbreviation only need to be defined once instead of 4 times

Line 252: “Study” should not be capitalized

Line 255: “Among 143 subjects… and 11.2% were SO” is unnecessary unless you want to delve more into the study details.

Line 256: The study states that only the sarcopenic obesity group was statistically significant. The obesity group was non-significantly increased. A larger sample size may increase the power to detect a significant difference in the obesity group

Line 264-266: “KD is… and cancer” would be better placed in the ketogenic diet section.

Line 267: “Study” should not be capitalized

Line 269: “VAT” abbreviation has already been defined

Line 270-274: why use this study on when there are studies directly looking at KD effect on inflammatory markers and/or body fat distribution? The study itself does not examine cytokines so it is inappropriate to say the results suggest “KD does not affect inflammatory cytokine production”. Also there are multiple grammatical errors that need to be corrected.

Line 276-277 NAFLD is an accumulation of fat in the liver with or without inflammation and fibrosis. Steatosis is just fat in the liver

Line 288: “Study” should not be capitalized

Line 292-295: this is only 1 sentence so should be added to another paragraph rather than being a stand-alone paragraph

Line 295: Parenthesis should be after “P < 0.001”

Line 303: “Ketogenic diets have been extensively studied in rodents” yet only 1 experiment from 2007 is cited. Please cite more extensive research or tone down the language

Line 304-306: discusses a study that mice on a ketogenic diet lose weight whereas a different study in lines 227-228 resulted in weight gain for mice on a ketogenic diet. These contradictory results should be discussed by the authors

Line 308: Remove “Regarding the NSFLD”

Line 310-311 “The added value… in liver” is not a complete sentence. Please fix the grammar

Line 312: Remove “This could be an added value, because”

Line 314: Remove “also”

Line 308-316: should the “rapid shift in gut microbiota composition” be mentioned in the gut microbiota section?

Line 317: remove “and fatigue” or add more content about fatigue

Line 326: “Study” should not be capitalized

Line 348-355: this new content has multiple grammatical errors that need to be revised

Line 364: “, which” should be between “mass” and “can”

Line 368-370: citations are not superscripted. “Whether” should be “While”. “Virtually… positive” is not a complete sentence

Author Response

Dear Reviewer, kindly find the attached file, please. 

Reviewer 3 Report

None of my comments from previous review were addressed. Please see the attached file.

Author Response

(The authors gave the same response as above.)

Round 3

Reviewer 2 Report

When I read the author's response to my comments, I was excited that they indicated extensive changes had been made to the manuscript. However, when I read the manuscript, I was disappointed that at least one of the indicated changes was not actually there. I am referring to the incorrect mechanism of ketone body generation provided and this needs to be corrected before even considering accepting the manuscript. I have annotated additional comments on the attached PDF. 

Author Response

I hope you are doing fine.

Kindly, find the attached file titled as “ Final – nutrients – 1412702 – manuscripts.”

The changes had been made according to review 3 and additional to review 2.

Reviewer 3 Report

The authors have improved the manuscript but need a few more changes. I have provided my comments in track changes. I suggest that they accept my edits and respond to my comments that are in red color.

Three are some major issues that must be addressed before it can be published.

Author Response

indly, find two attached files titled as following:

  1. Answers – nutrition – 1412702-review
  2. Version 3 – nutrition – 1412702 – manuscript

File (1), contains the answers to the review’s 3 questions. Whereas, File (2) is the corrected manuscript through review 3 comments. The corrected parts are all highlighted in red.
